# Fulminant MS Reactivation Following Combined Fingolimod Cessation and Yellow Fever Vaccination

**DOI:** 10.3390/ijms20235985

**Published:** 2019-11-28

**Authors:** Leoni Rolfes, Marc Pawlitzki, Steffen Pfeuffer, Christian Thomas, Jonas Schmidt-Chanasit, Catharina C. Gross, Andreas Schulte-Mecklenbeck, Heinz Wiendl, Sven G. Meuth, Oliver M. Grauer, Tobias Ruck

**Affiliations:** 1Department of Neurology with Institute of Translational Neurology, University Hospital Muenster, D-48149 Muenster, Germany; marc.pawlitzki@ukmuenster.de (M.P.); steffen.pfeuffer@ukmuenster.de (S.P.); catharina.gross@ukmuenster.de (C.C.G.); andreas.schulte-mecklenbeck@ukmuenster.de (A.S.-M.); heinz.wiendl@ukmuenster.de (H.W.); sven.meuth@ukmuenster.de (S.G.M.); oliver.grauer@ukmuenster.de (O.M.G.); tobias.ruck@ukmuenster.de (T.R.); 2Institute of Neuropathology, University Hospital Muenster, D-48149 Muenster, Germany; christian.thomas@ukmuenster.de; 3Bernhard Nocht Institute for Tropical Medicine, WHO Collaborating Centre for Arbovirus and Hemorrhagic Fever Reference and Research, D-20359 Hamburg, Germany; schmidt-chanasit@bnitm.de; 4Faculty of Mathematics, Informatics and Natural Sciences, University of Hamburg, D-20146 Hamburg, Germany

**Keywords:** fingolimod, multiple sclerosis, vaccination, yellow fever, autoimmune disease

## Abstract

A major concern caused by the discontinuation of disease modifying treatment for multiple sclerosis (MS) is a rebound of disease activity. Hypotheses about the underlying mechanism of fingolimod (FTY) induced exaggerated inflammatory responses are diverse. So far, vaccinations as a trigger for rebound activity following FTY suspension have not been described. However, several reports have highlighted the occurrence of neurological and autoimmune side effects after single or combined multi-vaccination procedures. Here, we describe the case of a highly active female MS patient demonstrating recurrent, severe MS relapses accompanied by extensive MRI activity, subsequent to yellow fever vaccination two months following FTY withdrawal. Blood and cerebrospinal fluid immunophenotyping indicated a B cell/plasma cell autoreactivity. Following a therapy with natalizumab the clinical, laboratory, MRI, and disease course improved significantly. This case hints towards a combined immunological mechanism characterized by molecular mimicry, bystander activation, and lymphocyte re-egress, resulting in extensive neurological impairment and shows that natalizumab represents a therapeutic option to counteract B cell mediated autoreactivity. Especially, the diagnostic and therapeutic management of this complex scenario might be instructive for clinical practice.

## 1. Introduction

Fingolimod (FTY) is an effective therapeutic option for treating relapsing-remitting multiple sclerosis (RRMS) [1]. Similar to other leukocyte-sequestering therapies such as natalizumab (NTZ), FTY withdrawal is associated with re-emerging disease activity and the formation of tumefactive lesions or an excessive inflammatory response similar to immune reconstitution inflammatory syndrome (IRIS) [2,3]. Exaggerated disease activity has also been reported following vaccination in the RRMS patient population [4].

Here, we present the case of an RRMS patient who developed recurrent, severe relapses following a yellow fever (YF) vaccination administered two months after FTY cessation. We outline histopathological analyses, radiological findings, and blood and cerebrospinal fluid (CSF) immunophenotyping to gain deeper insights into the immunological mechanisms underlying relapse development. Moreover, we discuss treatment strategies in this complex therapeutic scenario. A written informed consent was provided by the patient. The ethical approval for conduction was given by local ethical review board (University of Muenster, 2016-053-f-S; 05.04.2016).

## 2. Case Presentation

A 37-year-old, otherwise healthy female was diagnosed with RRMS in 2010. Due to ongoing disease activity, she was switched from glatiramer acetate treatment to FTY in 2014. Nonetheless, she presented with further relapses, accompanied by MRI activity, corresponding to an expanded disability status scale (EDSS) score of 1.5. In July 2018, she discontinued FTY due to travel plans and a related YF vaccination. Two months later, she presented with a sensorimotor hemiparesis (EDSS score: 4.5). Symptoms occurred one week after YF vaccination and were responsive to high-dose intravenous methylprednisolone (IVMPS) and plasma exchange (EDSS score at discharge: 2.0; Figure 1A).

Two months later, she presented with another relapse accompanied by tetraparesis, conjugated gaze palsy, aphasia, and dysphagia (EDSS score: 9.5). Cranial MRI showed massive, new and enlarged T2-FLAIR (Figure 1B) and gadolinium enhanced lesions. CSF analysis revealed a blood-brain barrier disturbance with lymphocytic pleocytosis and increased intrathecal immunoglobulin (Ig)G and IgM synthesis (Table 1, Figure 1C).

Interestingly, compared to previous findings from July 2018, we found new oligoclonal bands (OCB) (Figure 1D), indicating an activation of B cell/plasma cell mediated autoreactivity. Extensive pathogen and autoantibody diagnostics from CSF and serum were not conclusive. To ultimately exclude post-vaccinal YF encephalitis or progressive multifocal leukoencephalopathy, a brain biopsy (Figure 1E) with subsequent YF PCR was performed, and both were negative. Since repeated IVMPS treatment and immunoadsorption did not result in sufficient clinical improvement, the patient received NTZ (Figure 1A). Since then, clinical and MRI disease course improved significantly (EDSS score: 6.5, Figure 1A,B).

To detect distinct leukocyte subsets, we performed standardized multi-parameter flow cytometry of blood (following RBC lysis buffer) and CSF [5]. Samples were obtained under FTY therapy (T1), after the first infusion of NTZ (T2), and six months after NTZ initiation (T3) and were compared to respective FTY and NTZ treated control cohorts (Figure 1F,G). CSF B cell and plasma cell proportions at T2 were noticeably higher than proportions found at T1, T3, or in NTZ treated controls, supporting augmented B cell and plasma cell CNS autoreactivity at T2. Repeated courses of NTZ led to a decrease of both cell types (T3) (Figure 1F,G) and suppressed intrathecal Ig synthesis (Figure 1D). T cell or innate immune cell alterations were not observed.

## 3. Discussion

The inflammatory response boosted by FTY cessation is thought to be based on redistribution of immune cells, exacerbation of B cell mediated disease activity, and over-activation of the innate immune system [2]. Here, we report a severe rebound after FTY withdrawal due to further perturbation of the immune system by live vaccination.

Rebound disease activity after FTY cessation typically occurs within 4–16 weeks [6]. Further, post vaccination autoimmunity typically starts one week after immunization and occurs within a two-month risk period [7]. Thus, in light of the temporal relationship between events and the relapse severity, parallel occurrence of FTY withdrawal and YF vaccination may have led to fulminant disease activation, complicating diagnostic work-up and treatment decisions. The highly auto-reactive immune system induced severe and multifocal neurological deterioration accompanied by strong MRI activity suggestive of IRIS, as has been previously described following FTY discontinuation [3]. Of note, our CSF measures, showing increases in plasma cell and IgG/IgM synthesis and changes in OCB patterns, suggest a predominantly B cell driven pathology of exacerbated CNS inflammation.

Unfortunately, serological and CSF data shortly after FTY cessation as well as before and immediately after vaccination were not accessible. Thus, it is not definitely possible to distinguish which risk factor (FTY cessation vs. YF vaccination) played the major role and whether they interacted. Our predominately B-cell driven CSF pattern hints towards a key role of the YF vaccination in this case. This assumption is in line with the only one study on yellow fever vaccination in MS patients available [4]. This study included seven prospectively followed RRMS patients receiving YF vaccine. Notably, YF vaccination led to increased clinical and radiological deterioration within the same time period as in our case patient. However, in contrast to our scenario, all seven patients received DMTs with either interferon or glatiramer acetate, suggesting a lower pre-vaccination MS disease activity. In light of the evidence gained from this study and our case report, YF vaccines are not recommended in patients with MS [8,9,10].

## 4. Conclusions

In summary, our case indicates that (I) in the context of FTY withdrawal, MS relapses can be triggered by live attenuated vaccines, (II) a combined pathophysiological mechanism, characterized by molecular mimicry, bystander activation, and lymphocyte re-egress, can result in an atypical “IRIS-like” relapse phenotype, and (III) NTZ represents a valuable therapeutic option to counteract B cell mediated CNS autoreactivity immediately inhibiting lymphocyte transmigration into the CNS.

## Figures and Tables

**Figure 1 ijms-20-05985-f001:**
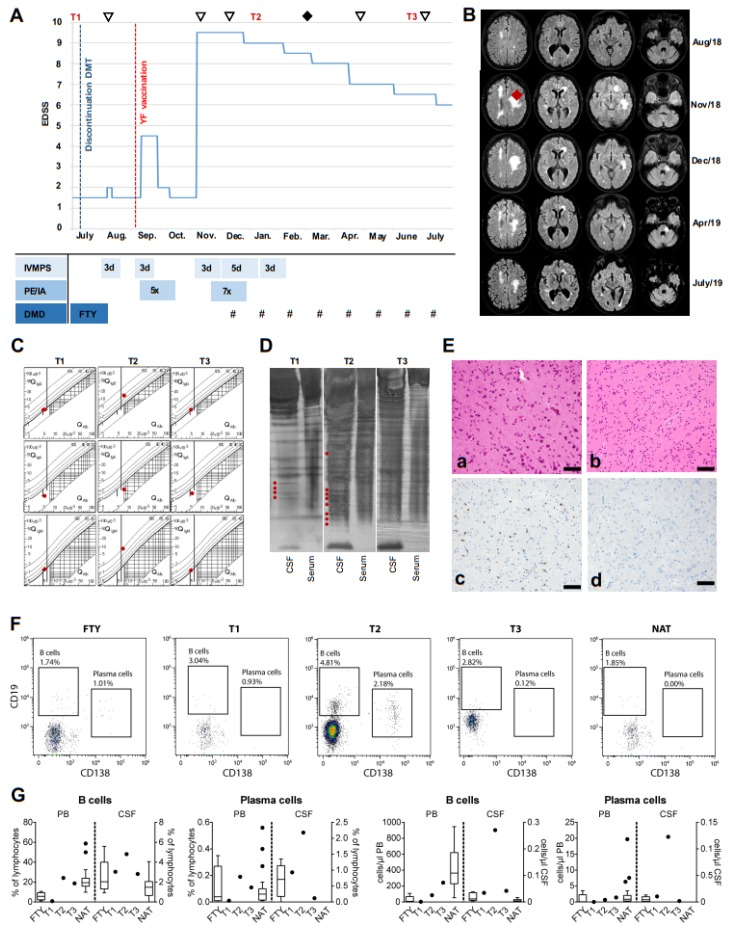
Time course of clinical relapse manifestation, diagnostics, and treatments in a RRMS patient following yellow fever (YF) vaccination after fingolimod cessation. (**A**) Chart shows the course of EDSS progression over time. Time points of additive diagnostic (MRI **▽**, biopsy♦) and CSF/Peripheral Blood Mononuclear Cell (PBMC) extraction (T1, T2, T3) are depicted. Bars present treatment regimens and duration of treatment in days (d, IVMPS = intravenous methylprednisolone, PE = plasma exchange, IA = immunoadsorption, DMD = disease modifying drug, FTY = fingolimod, # = natalizumab (NTZ)). (**B**) Panel displays MRI (T2-FLAIR) examinations at the time points illustrated in Figure 1A (**▽**). The localization of the brain biopsy is marked (♦). (**C**) CSF/serum quotient diagrams for IgG, IgA, and IgM, with hyperbolic graphs according to Reiber, showing an increased intrathecal IgG and IgM synthesis after vaccination and a normalization after recurrent NTZ infusions. Red points define patients’ values (**D**) OCB pattern changes after YF vaccination (red points illustrate increased OCB numbers in the CSF). (**E**) Upon histopathological examination mild reactive changes were visible in cortex (a) and white matter (b). Staining for CD68 demonstrated microgliosis (c), while staining for JC virus was negative (d). All scale bars denote 50 µm. (**F**) Dot plots of CSF analysis comparing representative FTY or NTZ treated patients to the YF patient (T1, T2, T3). (**G**) Proportions of CD19^+^ B cells and CD138^+^ plasma cells in PBMC (PB) and CSF. Cells were isolated from FTY or NTZ treated patients with RRMS (*n* = 5 for FTY, *n* = 28 for NTZ) and from the YF patient and analyzed by flow cytometry. Boxplots show median and 25% and 75% percentile; whiskers represent 5% and 95% percentile.

**Table 1 ijms-20-05985-t001:** CSF Analysis Results.

Variables	T1	T2	T3
Lymphocytes (cells/µL)	1	15	4
Protein (mg/L)	425	497	458
Q_Albumin_ (CSF/Serum)	5.1	7.6	5.5
BBB disturbance	no	yes	no
Q_IgG_ (CSF/Serum)	3.8	14.4	3.9
Intrathecal Synthesis (%)	0	60	0
Q_IgA_ (CSF/Serum)	1.3	2.4	1.6
Intrathecal Synthesis (%)	0	0	0
Q_IgM_ (CSF/Serum)	1.3	8.0	1.2
Intrathecal Synthesis (%)	21	77	
Oligoclonal bands	Type II	Type II	Type I
Q_Glucose_ (CSF/Serum)	0.51	0.53	0.56
Lactate (mmol/L)	1.88	2.34	1.83

Abbreviations: BBB (blood-brain barrier); CSF (cerebrospinal fluid). Pathological results are marked in red.

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
