# Peer review of "Fulminant MS Reactivation Following Combined Fingolimod Cessation and Yellow Fever Vaccination"

_ijms, 2019, doi:10.3390/ijms20235985_

Round 1
Reviewer 1 Report
Rolfes and colleagues reported on a case of severe multiple sclerosis re-activation after Fingolimod withdrawal and yellow fever vaccination. The case is elegantly studied and of great interest for the pathophysiology of MS re-activations and for the clinical practice. I have some minor suggestions to the authors.
For the clinical practice, it would be good knowing the level of lymphocytes in the peripheral blood at the time of the vaccination. I understand they might not be available, but it would be interesting to know whether disease re-activation occurred within lower lymphocyte levels, or normal.
In the discussion and/or conclusion, I would expand possible treatment consequences of this case. In light of the current treatment scenario, what disease modifying treatment would authors suggest to multiple sclerosis patients planning a travel requiring yellow fever vaccination? I believe patients with highly effective pulse therapy (e.g., cladribine) and normal lymphocyte levels could receive yellow fever vaccination. Would authors speculate the same? Or would authors safely not recommend yellow fever vaccination to multiple sclerosis patients?
Author Response
Comments from Reviewer 1:
‘For the clinical practice, it would be good knowing the level of lymphocytes in the peripheral blood at the time of the vaccination. I understand they might not be available, but it would be interesting to know whether disease re-activation occurred within lower lymphocyte levels, or normal.’
Response: We thank the reviewer for this important comment. We understand that the level of lymphocytes prior to vaccination could be an important feature to better understand the mechanism leading to this extensive disease activity. Unfortunately, the data is not available.
‘In the discussion and/or conclusion, I would expand possible treatment consequences of this case. In light of the current treatment scenario, what disease modifying treatment would authors suggest to multiple sclerosis patients planning a travel requiring yellow fever vaccination? I believe patients with highly effective pulse therapy (e.g., cladribine) and normal lymphocyte levels could receive yellow fever vaccination. Would authors speculate the same? Or would authors safely not recommend yellow fever vaccination to multiple sclerosis patients?’
Response: We thank the reviewer for this helpful comment. To date, rare data exist on YF vaccines in the context of highly active DMT, particularly for pulsed immune therapies. Cladribine as well as alemtuzumab promote a depletion with subsequent repopulation of T- and B-lymphocytes. A rebalancing of the immune system promoting immunoregulatory mechanisms is suggested as underlying mechanism for sustained disease control. Therefore, it is tempting to speculate that after lymphopenia an YF vaccination might be possible due to normalization of immune processes. However, the restoration of physiological immunity might not be complete and a vaccination might boost autoreactive immune responses leading to reoccurrence of disease activity. Even single autoreactive memory B or T lymphocytes might give rise to disease reactivation. As exaggerated disease activity after YF vaccination was also described in patients with low disease activity under ongoing baseline DMT, we generally do not recommend YF vaccination in patients in MS at all. To underline this recommendation and to address the reviewers’ comment in the manuscript we added the following statement: ‘This assumption is in line with the only one study on yellow fever vaccination in MS patients available [4]. This study included 7 prospectively followed RRMS patients receiving YF vaccine. Notably, YF vaccination led to increased clinical and radiological deterioration within the same time period as in our case patient. However, in contrast to our scenario, all 7 patients received DMTs with either interferon or glatiramer acetate, suggesting a lower pre-vaccination MS disease activity. In light of the evidence gained form this study and our case report, YF vaccines are not recommended in patients with MS [9, 10].’ (page 4, line 117-123).
Overall, we thank the reviewer for his/her constructive and very positive feedback.
Reviewer 2 Report
This in an interesting case study that highlights a number of issues in management of MS patients. The results are striking and the potential linkage to cessation of fingolimod and vaccination worthy of discussion. Clearly it is impossible to tell precisely what triggered the rebound in disease activity based on a single case, however the study would be more impactful if the authors were able to discuss related cases in more detail.
Author Response
Comments from Reviewer 2:
Clearly it is impossible to tell precisely what triggered the rebound in disease activity based on a single case, however the study would be more impactful if the authors were able to discuss related cases in more detail.
Response: Thank you for this important comment. We addressed this issue as stated above in a new paragraph in the discussion part (see response to reviewer 1 or page 7, line 3-10 in the manuscript):
Comments from Reviewer 1:
‘In the discussion and/or conclusion, I would expand possible treatment consequences of this case. In light of the current treatment scenario, what disease modifying treatment would authors suggest to multiple sclerosis patients planning a travel requiring yellow fever vaccination? I believe patients with highly effective pulse therapy (e.g., cladribine) and normal lymphocyte levels could receive yellow fever vaccination. Would authors speculate the same? Or would authors safely not recommend yellow fever vaccination to multiple sclerosis patients?’
Response: [...] ‘This assumption is in line with the only one study on yellow fever vaccination in MS patients available [4]. This study included 7 prospectively followed RRMS patients receiving YF vaccine. Notably, YF vaccination led to increased clinical and radiological deterioration within the same time period as in our case patient. However, in contrast to our scenario, all 7 patients received DMTs with either interferon or glatiramer acetate, suggesting a lower pre-vaccination MS disease activity. In light of the evidence gained form this study and our case report, YF vaccines are not recommended in patients with MS [9, 10].’ (page 4, line 117-123).